# Water production efficiency and economic benefits under diversified planting modes of intercropping-multiple cropping in arid regions

**Na Zhang**[1,2], **Jianxin Jin**[3], **Jing Chen**[2]*

**1** Ningxia Hui Autonomous Region Water Conservancy Research Institute, Yinchuan, China, **2** Hohai University, Nanjing, China, **3** Institute of Agricultural Resources and Environment, Ningxia Academy of Agricultural and Forestry Sciences, Yinchuan, China

* 2156640980@qq.com

## Abstract

Diversified planting patterns are important measures to improve the comprehensive grain production capacity, alleviate the contradiction between grain crops and forage planting, and enhance water production efficiency. In order to explore the suitable diversified planting modes in the Yellow River irrigation area of Ningxia, a total of 4 treatments were designed, including wheat-maize silage intercropping and multiple planting of sorghum-sudangrass hybrid after wheat harvest (T1); wheat-cabbage intercropping, multiple planting of oil sunflower after wheat harvest, followed by maize silage planting after cabbage harvest (T2); sole wheat, after the harvest of wheat, half of the land is used for multiple plantings of maize silage, and the other half is used for multiple plantings of oil sunflower (T3); and sole maize silage (T4). The results showed that all diversified planting modes can increase biomass and land equivalent ratio compared to the control monoculture. The T2 had the highest total biomass and land equivalent ratio, the two-year average was 79.68 t/ha and 2.27, respectively. The highest biological yield per cubic meter of water was achieved by T3, with an average of 12.65 kg/m³ over two years. T1 achieved the highest output value per cubic meter of water, with 16.81 Chinese Yuan/m³ over 2 years. In both T1 and T2, due to the increased spacing between intercropping crops, as well as sufficient water and fertilizer supply and developed root system, maize silage is at a competitive advantage, with a interspecific relative competitive ability between 0.059–0.234. When maize silage and oil sunflower are planted simultaneously, due to the fast growth rate of oil sunflower, it is the dominant crop, the interspecific relative competitive ability in 2022 and 2023 were 0.164 and 0.137, respectively. The net benefit of T1 was the highest, with an average total net income of 84,950 Chinese Yuan/ha over 2 years. It can be seen that diversified planting patterns can improve the yield and economic benefits per unit land area, and are a highly promising planting pattern.

**Data availability statement:** All relevant data are within the article and its Supporting Information files.

**Funding:** This work was supported by the National Key Research & Development Program of China (grant numbers 2021YFD1900600 and 2022YFG1900205, awarded to Dr. Na Zhang), the Key Research & Development Program of Ningxia Hui Autonomous Region (grant number 2021BEF02034, awarded to Jianxin Jin), and the Ningxia Natural Science Foundation Project (grant number 2022AAC03725, awarded to Dr. Na Zhang).

**Competing interests:** The authors have declared that no competing interests exist.

## Introduction

The shortage of water resources and uneven spatial and temporal distribution are currently the main limiting factors for the development of agricultural production in northwest China [1]. The Ningxia Yellow River Irrigation Area is located in the arid and water deficient northwest plateau, the region has low rainfall and high evaporation, and agricultural water mainly relies on the Yellow River for water diversion [2]. In recent years, with the decreased water inflow from the Yellow River and the restricted water usage indicators, along with the rapid expansion of animal husbandry, have exacerbated the competition for land, water and other resources between grain crops and feed crops, the expected shortfalls in land area and water are anticipated to exceed 3,000 hectares and one billion cubic meters, respectively, in 2025 [3–4]. At that juncture, it will constitute a significant threat to regional food security and human health. In order to increase the unit land area and unit water resource output, achieve efficient and sustainable utilization of water resources in the Yellow River irrigation area of Ningxia, ensure the safety of food and forage crop supply, and the sustainable development of agriculture, The ancient planting mode of intercropping and multiple cropping has once again attracted widespread attention from local scholars [5].

The intercropping and multiple cropping, as a long-standing and efficient intensive planting method, improves the utilization efficiency of water, nutrients, light energy and other resources through the reasonable combination of different crops in time and space [6–7]. However, under the intercropping model of two crops, sowing width, planting density, and variety combination have a significant impact on the crop growth of the intercropping system [8]. Currently, many scholars have explored suitable planting ratios and bands for intercropping systems such as maize-common bean [9], maize-black gram, green gram, vegetable cowpea, sea, ground nut [10], maize-canopy [11], as well as optimized fertilization under intercropping modes [12–13], intercropping advantages [14] and resource utilization of intercropping between two crops [15]. Various suitable intercropping planting modes and water and fertilizer management schemes have been proposed [16]. There are also a lot of studies on multiple cropping, which has increased the sown area, improved grain yield, and effectively enhanced grain production capacity of land [17]. Many multiple cropping modes have been proposed based on temperature and crop characteristics, such as multiple cropping vegetables after wheat [18] and multiple cropping grass after wheat [19]. Significant achievements have also been made in water use in intercropping cultivation modes of multiple crops. Many research has found that under intercropping modes, there are two aspects of competition and complementarity in water use between two crops, and this relationship is mainly influenced by various factors such as crop types, irrigation and fertilization systems, planting density, spatial layout, tillage and coverage measures, and environmental factors [15,20]. The crop variety has a significant impact on the water use efficiency of intercropping systems [21]. In the soybean-wheat, maize system, the soil moisture in soybean strips can be used by intercropping wheat or maize, increasing the soil water storage capacity of wheat and maize strips and improving the soil water use efficiency [22]. In the wheat-maize intercropping system, wheat has stronger water competitiveness. Although the total water consumption of the intercropping system has decreased compared to monoculture, this is mainly due to the reduction in water consumption of wheat [23]. The research on diversified planting systems, including intercropping and multiple cropping, within the Yellow River irrigation region of Ningxia, predominantly focuses on the intercropping of maize and soybean, as well as multiple cropping following wheat harvest, explored the characteristics of crop physiology, yield, and comprehensive benefits under different planting systems [24,25].

However, it can be seen that current research on intercropping and multiple cropping is still conducted separately, and the research objects are relatively single. There are few reports

on the composite diversified planting modes involving various crops such as grain crops, forage crops, and vegetables, and there is a lack of research on the advantages of intercropping and water use efficiency under intercropping and multiple cropping systems in the same year. Therefore, in order to screen for intercropping and multiple cropping planting modes that are suitable for the Yellow River irrigation area in Ningxia and can simultaneously produce grain crops and forage crops, with high water production efficiency and planting benefits. This study investigates the intercropping advantages, water production efficiency, competitive advantages, and economic benefits of different intercropping and multiple cropping systems for grain crops such as wheat and maize, as well as other forage crops and vegetables such as cabbage, sorghum-sudangrass hybrid, and maize silage. Suitable intercropping and multiple planting modes and crop species are proposed for the Yellow River Irrigation Area in Ningxia and other similar ecological regions worldwide, thus providing a scientific basis for the promotion of diversified planting modes.

## Materials and methods

### Overview of the experimental area

The experimental base is located in Wanghong Town, Yongning County, China, with a geographical location of 37°55′ N latitude, 106°6′ E longitude, and an altitude of 1106 m. It has a typical temperate continental climate with abundant solar and thermal resources. The average annual temperature is 9.3°C, the annual sunshine hours are 2974.4 h, the average frost free period is 171 d, the average annual precipitation is 260.7 mm, and the average annual evaporation is 2013.7 mm. The soil in the experimental area is sticky loam soil, with an average field water holding capacity of 33.54% (volume percentage) and an average soil bulk density of 1.37 g/cm³. Alkaline hydrolyzed nitrogen is 82 mg/kg, available phosphorus is 26.2 mg/kg, available potassium is 147 mg/kg, and organic matter content is 9.2 g/kg. The location of the experimental area and test photos are shown in Fig 1.

### Experimental design

In intercropping and multiple cropping modes, to improve water use efficiency, it is necessary to increase the planting of maize, and in order to increase dry matter yield per unit area, Sorghum-sudangrass hybrid should be replaced by oil sunflower for replanting, the economic benefits should be analyzed based on the market prices of forage and vegetables. Therefore, a total of 4 diversified planting modes were designed, including wheat-maize silage intercropping and multiple planting of sorghum-sudangrass hybrid after wheat harvest (T1). The wheat strip width is 2.2 m and is planted using mechanized uniform sowing. There are 4 rows of maize silage with a row spacing of 0.5 m, and the edge row distance between wheat and maize silage is 0.3 m. Planting 4 rows of sorghum-sudangrass hybrid, with a wide row spacing of 0.7 m and a narrow row spacing of 0.4 m, and a distance of 0.7 m from the edge row to the maize silage. The area of wheat and sorghum-sudangrass hybrid plots was 0.0367 ha, and for maize silage it was 0.03 ha. Wheat-cabbage intercropping, multiple planting of oil sunflower after wheat harvest, followed by maize silage planting after cabbage harvest (T2). Cabbage was planted in ridges, with a total of 2 ridges and 4 rows, with a ridge spacing of 0.4 m and a row spacing of 0.3 m. Oil sunflower planting in 6 rows, with a row spacing of 0.4 m, and the distance between oil sunflower and maize silage is 0.4 m, the area of wheat and oil sunflower plots was 0.0367 ha, and the area for cabbage and maize silage was 0.03 ha; sole wheat, after the harvest of wheat, half of the land is used for multiple plantings of maize silage, and the other half is used for multiple plantings of oil sunflower (T3), planting method is the same as T2, the area of wheat plots was 0.0667 ha, and the area for maize silage and oil sunflower was 0.03385 ha; sole maize silage

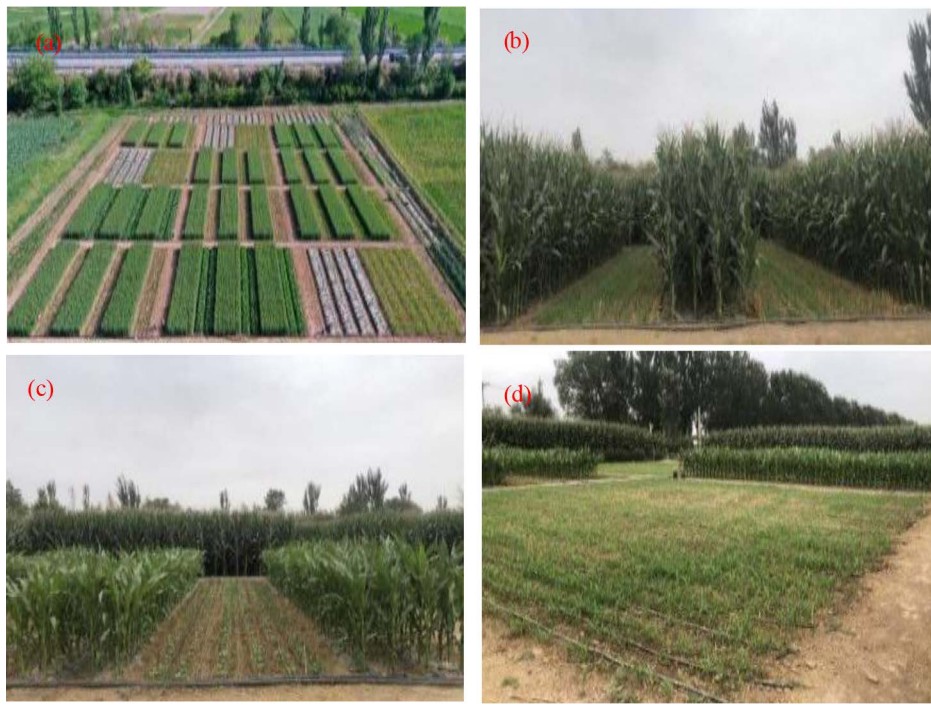

**Fig 1. Images from different experimental treatments in the experimental site.** A is the overall view of the experimental base, b is intercropping wheat with maize silage and replanting sorghum-sudangrass hybrid after wheat harvest, c is intercropping wheat with cabbage, replanting oil sunflower after wheat harvest, and replanting maize silage after cabbage harvest, d is sole wheat, after wheat harvest, half of the land is replanted with maize silage, and half of the land is replanted with oil sunflower. Reprinted from [ref] under a CC BY license, with permission from [name of publisher], original copyright [original copyright year].

(T4), planting with equal row spacing, row spacing of 0.5 m, the area of plots was 0.0667 ha. Each treatment set has 3 repetitions, and each plot has an area of 0.0667 ha. Except for T4, the irrigation method is drip irrigation, with a drip head spacing of 0.3 m and a drip head flow rate of 2 L/h. The distance between drip irrigation belts for wheat is 0.5 m, and other crops are arranged in a row along the belt. The sowing date, harvesting date, and irrigation scheduling of each crop for different experimental treatments are shown in Table 1, and the diagrammatic framework and evaluation index of experimental protocol are shown in Fig 2.

## Indicators and methods

**Observation indicators and methods.** (1) Biomass: Harvest 1 m² of fresh ground samples of each crop during the harvest period and bring them back to the laboratory. Use an oven to first adjust to 105 °C for wilting and then adjust to 80 °C for drying until a constant weight is achieved.

(2) Soil moisture content: Determine the soil moisture content before and after each irrigation and after rainfall using the drying and weighing method. The sampling location is at the edge row and 20 cm inward for wheat, and at the edge row and one inward row for other crops.

**Evaluating indicator.** Intercropping advantage: The intercropping advantage of intercropping populations is usually measured by two indicators: land equivalent ratio (LER),

**Table 1. The sowing date, harvesting date, and irrigation scheduling of each crop for different experimental treatments. The irrigation amount and irrigation frequency of each crop in 2022 and 2023 will remain unchanged, and the sowing and harvesting dates will be kept within two days for two years.**

| Treatments | Crop | Planting type | Sowing date | Harvest date | Irrigation | |
|---|---|---|---|---|---|---|
| | | | | | Amount/(mm) | Frequency/(time) |
| T1 | Wheat | Normal planting | March 2nd | July 10th | 285 | 6 |
| | Maize silage | Normal planting | April 15th | September 17th | 277.5 | 8 |
| | Sorghum-sudangrass hybrid | Multiple cropping | July 13th | September 17th | 120 | 3 |
| T2 | Wheat | Normal planting | March 2nd | July 10th | 285 | 6 |
| | Cabbage | Normal planting | March 24th | June 2nd | 260 | 7 |
| | Oil sunflower | Multiple cropping of wheat | July 13th | September 17th | 120 | 3 |
| | Maize silage | Multiple cropping of cabbage | June 5nd | September 17th | 207 | 6 |
| T3 | Wheat | Normal planting | March 2nd | July 10th | 285 | 6 |
| | Maize silage | Multiple cropping | July 13th | September 17th | 120 | 3 |
| | Oil sunflower | Multiple cropping | July 14th | September 17th | 120 | 3 |
| T4 | Maize silage | Normal planting | April 15th | September 17th | 520 | 5 |

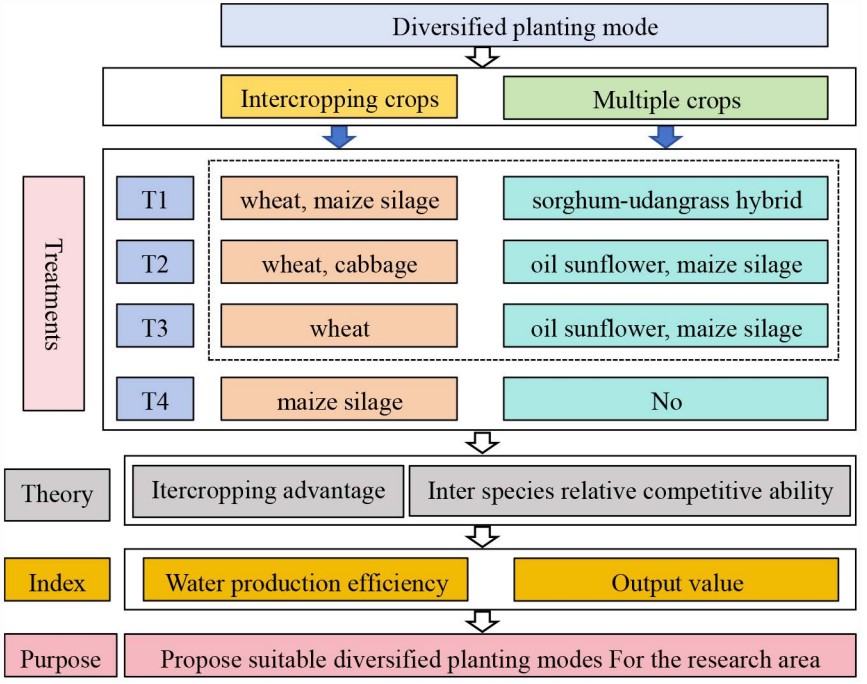

**Fig 2. The diagrammatic framework and evaluation index of experimental protocol.**

which represents the difference between the yield of intercropping populations and the yield of monoculture with the same crop and area [26]. The calculation formula for LER is shown in (1-1):

$$P_{LER} = P_{LER_1} + P_{LER_2} + \ldots = I_1 / Y_1 + I_2 / Y_2 + \ldots \tag{1-1}$$

Where: $P_{LER}$ is land equivalence ratio of intercropping groups, $P_{LER_1}$ is the relative yield of first crop in intercropping, $P_{LER_2}$ is relative yield of second crop in intercropping, relative

yield is also known as the partial land equivalent ratio of a certain crop in an intercropping population. $I_1$ and $Y_1$ are the yield of a crop in an intercropping group and its corresponding monoculture yield, $I_2$ and $Y_2$ is the yield of another crop in the intercropping group and its corresponding monoculture yield, respectively. The formula for calculating the difference in yield between intercropping populations and monoculture with the same area of corresponding crops is:

$$Y = Y_S - \left(Y_1\alpha + Y_2\beta + ...\right) \qquad (1\text{-}2)$$

Where: $Y$ is intercropping population biomass yield advantage, t/ha; $Y_S$ is the sum of the biomass yields of two crops in an intercropping group, t/ha; and α and β are the proportion of each crop area in the entire planting area.

Interspecific relative competitive ability: Evaluating the competitive ability of one crop in intercropping populations relative to the other for growth resources such as water, fertilizer, air, and heat, we use interspecific relative competitiveness, calculate using equations (1-3):

$$A_{1,2} = I_1 / \left(Y_1\alpha\right) - I_2 / \left(Y_2\alpha\right) \qquad (1\text{-}3)$$

$A_{1,2} > 0$, indicating that the competitiveness of the first crop in the intercropping population is greater than that of the second crop; $A_{1,2} < 0$, indicating that the competitiveness of the second crop in the intercropping population is greater than that of the first crop.

## Results

### Intercropping advantage

Advantages of intercropping under different intercropping and multiple cropping modes are shown in Table 2. Analysis shows that under intercropping and multiple cropping modes, all T1–T3 treatments increased the land equivalence ratio, with significant differences between the treatments. Among them, T2 reached the highest level, followed by T3 and T1. T2 was 11.01% larger than T3 and 28.25% larger than T1. Specifically, the T2 treatment achieved $P_{LER}$ of 2.24 and 2.30 in 2022 and 2023, respectively. This is because the T2 treatment planted four crops, including maize silage, a crop with high water production efficiency, which greatly increased the yield of ground biomass compared to sole cropping. Next is the T3 treatment, with a $P_{LER}$ of 2.04 and 2.05 in 2022 and 2023, respectively. This model involves intercropping oil sunflower and maize silage after wheat harvest, which greatly increases biological yield compared to maize monoculture. Among the three intercropping modes, the T1 treatment had the smallest $P_{LER}$ of 1.76 and 1.78 in 2022 and 2023, respectively, but it also greatly increased the biomass yield per unit area compared to maize monoculture. Among all treatments, T2 had the highest total biomass, followed by T3 and T1, T4 was the smallest. The total biomass yield of the T1 treatment increased by 9.9 t/ha and 9.3 t/ha compared to the same area of maize monoculture, an increase of 25.29% and 23.48% in 2022 and 2023, respectively. The increase in total biomass was most significant in the T2 treatment, with an increase of 39.73 t/ha and 40.93 t/ha compared to maize monoculture, an increase of 101.53% and 104.48%, respectively, in the 2 years. The T3 treatment also demonstrated a significant advantage in increasing total biomass, with an increase of 17.39 t/ha and 17.99 t/ha compared to monoculture maize, an increase of 44.44% and 45.45%. From the perspective of intercropping population biomass yield advantage, the T2 and T1 treatments have significant advantages in biomass increase compared to monocropping on the same land area, with $Y$ values of 8.25 t/ha and 7.35 t/ha in 2022, and $Y$ values of 9.75 t/ha and 7.05 t/ha in 2023. However,

**Table 2. Advantages of intercropping under different intercropping and multiple cropping modes in 2022-2023.** The biomass of each crop is measured at harvest, with a sample size of n = 9, establish three plots for each treatment, and within each plot, randomly select three sampling points. The standard deviation of wheat biomass for each treatment ranges from 0.08 to 0.13. For maize silage, it ranges from 0.31 to 0.49. For the sorghum-sudangrass hybrid, it is between 0.36 and 0.44. For cabbage, it ranges from 0.39 to 0.59. For oil sunflower, it is between 0.40 and 0.56.

| Years | Modes | Crops | Cultivated area/(ha) | Biomass/(t/ha) | $P_{LERi}$ | $P_{LER}$ | Y/(t/ha) |
|---|---|---|---|---|---|---|---|
| 2022 | T1 | Wheat | 0.0367 | 17.44 ± 3.27 | 0.57 ± 0.05 | 1.76 ± 0.25[Cc] | 7.35 ± 1.14[Bb] |
| | | Maize silage | 0.03 | 51.67 ± 6.99 | 0.59 ± 0.06 | | |
| | | Sorghum-sudangrass hybrid | 0.0367 | 29.43 ± 5.18 | 0.60 ± 0.12 | | |
| | T2 | Wheat | 0.0367 | 19.35 ± 2.45 | 0.63 ± 0.08 | 2.24 ± 0.18[Aa] | 8.25 ± 0.95[Aa] |
| | | Cabbage | 0.03 | 57.99 ± 5.67 | 0.54 ± 0.11 | | |
| | | Maize silage | 0.03 | 40.99 ± 6.99 | 0.47 ± 0.07 | | |
| | | Oil sunflower | 0.0367 | 43.05 ± 3.81 | 0.60 ± 0.12 | | |
| | T3 | Wheat | 0.0667 | 17.69 ± 1.19 | 1.06 ± 0.29 | 2.04 ± 0.12[Bb] | −0.30 ± 0.08[Cc] |
| | | Maize silage | 0.03385 | 41.36 ± 6.79 | 0.53 ± 0.13 | | |
| | | Oil sunflower | 0.03385 | 35.16 ± 4.73 | 0.45 ± 0.15 | | |
| | T4 | Maize silage | 0.0667 | 39.13 ± 1.79 | | | |
| 2023 | T1 | Wheat | 0.0367 | 17.98 ± 2.18 | 0.59 ± 0.05 | 1.78 ± 0.18[Cc] | 7.05 ± 1.24[Bb] |
| | | Maize silage | 0.03 | 49.33 ± 3.67 | 0.56 ± 0.04 | | |
| | | Sorghum-sudangrass hybrid | 0.0367 | 30.52 ± 3.81 | 0.63 ± 0.11 | | |
| | T2 | Wheat | 0.0367 | 20.71 ± 2.45 | 0.68 ± 0.06 | 2.30 ± 0.22[Aa] | 9.75 ± 1.22[Aa] |
| | | Cabbage | 0.03 | 59.33 ± 6.99 | 0.55 ± 0.05 | | |
| | | Maize silage | 0.03 | 40.67 ± 3.99 | 0.46 ± 0.08 | | |
| | | Oil sunflower | 0.0367 | 43.87 ± 2.18 | 0.61 ± 0.14 | | |
| | T3 | Wheat | 0.0667 | 17.39 ± 1.05 | 1.03 ± 0.15 | 2.05 ± 0.19[Bb] | 0.45 ± 0.07[Cc] |
| | | Maize silage | 0.03385 | 42.25 ± 3.25 | 0.54 ± 0.09 | | |
| | | Oil sunflower | 0.03385 | 36.93 ± 2.66 | 0.47 ± 0.05 | | |
| | T4 | Maize silage | 0.0667 | 39.58 ± 3.45 | | | |

Note: The results of comparing the land equivalent ratio in the table with that of a sole maize variety, where uppercase letters indicate 5% significance and lowercase letters indicate 1% significance.

the T3 treatment showed no significant pattern, decreasing by 0.3 t/ha in 2022 and increasing by 0.45 t/ha in 2023, indicating a certain degree of variability. It can be seen that through intercropping and multiple cropping, the total biomass yield per unit area can be significantly increased, and the biodiversity of farmland is also increased, with obvious intercropping advantages and yield-increasing effects.

## Water production efficiency

To evaluate the biological yield and economic benefits of water input per unit, the biological yield per cubic meter of water and the output value per cubic meter of water were used to analyze and evaluate the water production benefits under the intercropping-multiple cropping diversified planting model (Fig 3). The analysis shows that there are significant differences in biomass and output value generated per unit of water input under different planting modes. The T3 had the highest biological yield per cubic meter of water, reaching 12.6 kg/m³ in 2022 and 12.7 kg/m³ in 2023. It was followed by T1 and T2, while the T4 had the lowest value. T3 was 3.25%–4.13% larger than T1, 30.16%–30.79% larger than T2 and 144.71%–145.14% larger than T4 from 2022 to 2023. This indicates that intercropping and multiple cropping planting modes, combined with efficient water-saving irrigation, significantly improve the production capacity of biomass yield per cubic meter of water. Another indicator for evaluating the

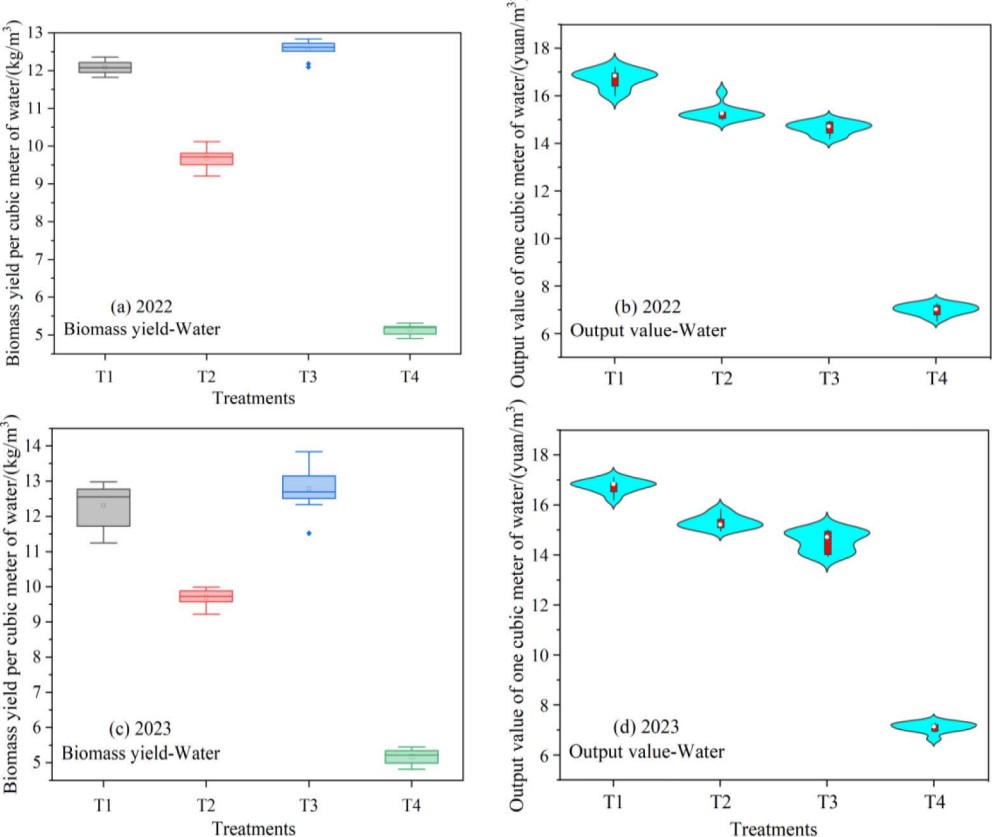

**Fig 3. The biological yield per cubic meter of water (a) and output value per cubic meter of water (b) in 2022-2023.** The sample size for each treatment is n = 9, establish three plots for each treatment, and within each plot, randomly select three sampling points. The standard deviations of the biomass yield per cubic meter of water, averaged over the two years for T1 to T4, were 0.59, 0.49, 0.52, and 0.45, respectively. The standard deviations of the average output value of one cubic meter of water treated by T1-T4 over two years are 0.41, 0.29, 0.32, and 0.33, respectively.

superiority of water use efficiency in intercropping and multiple cropping models is the output value per cubic meter of water, which is also an important factor affecting the promotion of diversified planting models. The T1 had the highest output value per cubic meter of water, reaching 16.73 Chinese Yuan/m³ in 2022 and 16.88 Chinese Yuan/m³ in 2023. It was followed by T2 and T3, while the T4 had the lowest output value, T3 was 8.64%–8.73% larger than T1, 14.28%–15.63% larger than T2 and 136.15%–139.00% larger than T4 from 2022 to 2023. This was mainly because the T4 used flood irrigation, which required a larger amount of water, and only one crop of maize silage was planted, resulting in a decrease in the output value of one cubic meter of water. On the other hand, T1 had a higher output value due to both grain and grass production, and it has a smaller irrigation amount.

## Interspecific relative competitive ability

When two crops are intercropped, different crops compete for water, fertilizer, and light and heat resources. Due to differences in sowing time, root distribution, and aboveground growth, the two crops may exhibit competitive advantages or disadvantages, which can sometimes be reversed. The interspecific relative competitive ability of two intercropping crops under different planting modes is shown in Fig 4. Compared to wheat and the multiple cropping of

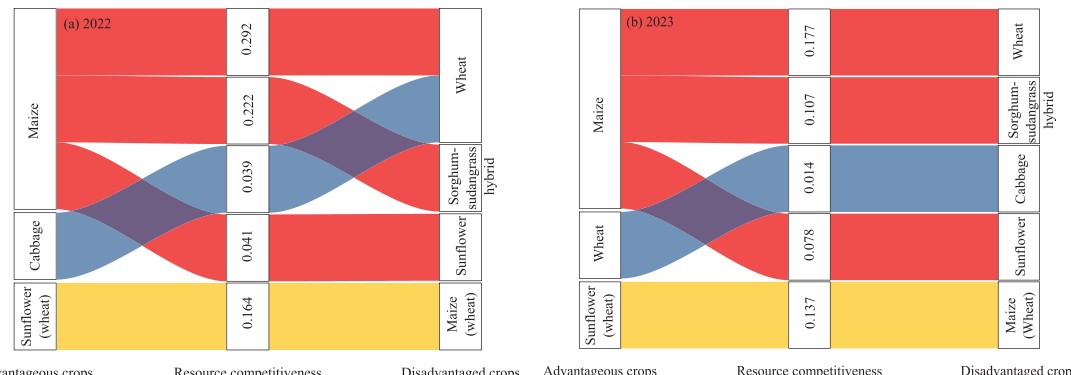

**Fig 4. The interspecific relative competitive ability of two intercropping crops in 2022(a) and 2023 (b), Sunflowers (wheat) and maize (wheat) were replanted after T3 treatment with wheat.** The sample size for each treatment is n = 9, establish three plots for each treatment, and within each plot, randomly select three sampling points.

sorghum-sudangrass hybrid after wheat, maize silage has a competitive advantage in terms of the final biological yield, the interspecific relative competitive ability is 0.292 and 0.222 in 2022, 0.177 and 0.107 in 2023, respectively. Under this cultivation mode, the increased distance between maize silage and wheat, along with maize silage's wide root distribution and taller plant height, enables it to better access growth resources both above and below ground. The cabbage-wheat intercropping cropping pattern, there was no significant trend. The cabbage showed a competitive advantage in 2022, the interspecific relative competitive ability was 0.039, while wheat showed a competitive advantage in 2023, the interspecific relative competitive ability was 0.014. When maize silage and sunflower were intercropped, maize silage showed a competitive advantage over sunflower for resources in T2 treatment, with relative interspecific competitive abilities of 0.041 in 2022 and 0.078 in 2023, respectively. Due to the shorter growth period of cabbage and earlier sowing of maize silage in multiple cropping, the resource competitiveness of maize silage is greater than that of sunflower. However, when maize silage and oil sunflower were sown simultaneously (T3), sunflower showed a competitive advantage in resources, with interspecific relative competitive ability of 0.164 in 2022 and 0.137 in 2023. This was mainly because the growth period of the oil sunflower was short, the growth rate was fast, and the root system and aboveground parts could occupy a dominant position quickly compared to maize silage. In summary, when maize silage is intercropped with other crops, it generally exhibits a competitive advantage in resource acquisition. Both cabbage and wheat have similar competitive abilities for resources, so it is necessary to reasonably match the crop varieties within the intercropping population based on available water and fertilizer resources.

## Output value of each mode

Due to differences in market demand, the economic efficiency of different planting modes will also show significant differences (Table 3). Analysis shows that T1 has the highest net benefit, with a total net income of 84,800 Chinese Yuan/ha for all crops in 2022, and 85,100 Chinese Yuan/ha in 2023, followed by T2 and T3, T4 has the smallest net benefit. T1-T3 treatments have shown an increase of 78.51%–178.51% in 2022 and 78.26%–146.67% in 2023, respectively, in comparison to T4 (control). Among all crops, maize silage has the highest net benefit, with the net income reaching 2,430–57,600 Chinese Yuan/ha within two years in each planting mode, and the highest net benefit is in T1 treatment, because maize silage has

**Table 3. Economic benefits of different planting modes in 2022-2023.** The sample size for each treatment is n = 9, establish three plots for each treatment, and within each plot, randomly select three sampling points. The standard deviation of net income for each planting mode in 2022 ranges from 0.33 to 0.41, while in 2023 it ranges from 0.28 to 0.39.

| Years | Modes | Crops | Output value of various crops/ (ten thousand Chinese yuan/ha) | Input value of various crops/ (ten thousand Chinese yuan/ha) | Net income for each mode/(ten thousand Chinese Yuan/ha) |
|---|---|---|---|---|---|
| 2022 | T1 | Wheat | 3.20 ± 0.12 | 1.23 ± 0.08 | 8.48 ± 0.22[Bb] |
| | | Maize silage | 7.32 ± 0.32 | 1.56 ± 0.11 | |
| | | Gaodan grass | 1.38 ± 0.09 | 0.63 ± 0.05 | |
| | T2 | Wheat | 3.51 ± 0.31 | 1.25 ± 0.09 | 8.39 ± 0.26[Aa] |
| | | Cabbage | 3.60 ± 0.28 | 2.33 ± 0.13 | |
| | | Maize silage | 4.23 ± 0.38 | 0.90 ± 0.18 | |
| | | Oil sunflower | 2.16 ± 0.16 | 0.63 ± 0.12 | |
| | T3 | Wheat | 3.03 ± 0.11 | 1.24 ± 0.09 | 6.23 ± 0.18[Cc] |
| | | Maize silage | 3.07 ± 0.23 | 0.64 ± 0.10 | |
| | | Oil sunflower | 2.65 ± 0.17 | 0.64 ± 0.08 | |
| | T4 | Maize silage | 5.01 ± 0.22 | 1.52 ± 0.22 | 3.49 ± 0.11[Dd] |
| 2023 | T1 | Wheat | 3.24 ± 0.16 | 1.25 ± 0.07 | 8.51 ± 0.28[Bb] |
| | | Maize silage | 7.41 ± 0.24 | 1.62 ± 0.09 | |
| | | Gaodan grass | 1.35 ± 0.15 | 0.64 ± 0.12 | |
| | T2 | Wheat | 3.54 ± 0.22 | 1.23 ± 0.08 | 8.43 ± 0.31[Aa] |
| | | Cabbage | 3.58 ± 0.18 | 2.37 ± 0.17 | |
| | | Maize silage | 2.33 ± 0.11 | 0.95 ± 0.08 | |
| | | Oil sunflower | 2.21 ± 0.16 | 0.68 ± 0.13 | |
| | T3 | Wheat | 2.98 ± 0.21 | 1.27 ± 0.11 | 6.15 ± 0.22[Cc] |
| | | Maize silage | 3.14 ± 0.24 | 0.68 ± 0.07 | |
| | | Oil sunflower | 2.64 ± 0.18 | 0.66 ± 0.15 | |
| | T4 | Maize silage | 4.98 ± 0.19 | 1.53 ± 0.19 | 3.45 ± 0.14[Dd] |

Note: The results of comparing the land equivalent ratio in the table with that of a sole maize variety, where uppercase letters indicate 5% significance and lowercase letters indicate 1% significance.

significant competitive advantages over wheat and sorghum-sudangrass hybrid under T1 treatment. At the same time, maize silage will harvest both grains and straw in this mode, thus having high economic benefits. Among wheat cultivated under various modes, T3 treatment has the lowest economic benefits with a net income of only 17,900 Chinese Yuan/ha in 2022 and 17,100 Chinese Yuan/ha in 2023, followed by T1 treatment, with 19,700 Chinese Yuan/ha and 19,900 Chinese Yuan/ha for two years respectively. The T2 treatment has the highest net income for wheat, amounting to 22,600 Chinese Yuan/ha and 23,100 Chinese Yuan/ha over the course of two years, respectively. This may be related to the variety of wheat used in intercropping. Compared with maize silage, wheat has a stronger interspecific competition ability when grown with cabbage. Among different crops, the economic benefits of sorghum-sudangrass hybrid and cabbage are the lowest. Specifically, sorghum-sudangrass hybrid had a net income of 7,500 Chinese Yuan/ha in 2022 and 7,100 Chinese Yuan/ha in 2023, while cabbage had a net income of 12,700 Chinese Yuan/ha in 2022 and 12,100 Chinese Yuan/ha in 2023. This is mainly due to the lower yield of sorghum-sudangrass hybrid and the higher input for cabbage, resulting in lower net incomes for these two crops. In summary, in intercropping and multiple cropping modes, due to the water replenishment relationships between different crops, they can improve economic benefits to a certain extent compared to monoculture, making these modes potential cultivation options.

## Discussion

This study explored the performance of different diversified planting patterns in terms of total biomass, land equivalent ratio, water use efficiency, interspecific relative competitive ability, and output value by setting up different intercropping and multiple cropping treatments. It was found that the treatment of wheat-cabbage intercropping, with multiple planting of oil sunflower after wheat harvest, followed by maize silage planting after cabbage harvest had advantages in total biomass and land equivalent ratio. The treatment of sole wheat, after the harvest of wheat, half of the land was used for multiple plantings of maize silage, and the other half was used for multiple plantings of oil sunflower had the highest biological yield per cubic meter of water. The treatment of wheat-maize silage intercropping and multiple planting of sorghum-sudangrass hybrid after wheat harvest had the highest output value per cubic meter of water and the highest net benefit. The amount of heat available in a region is the key of intercropping and multiple cropping patterns, and also determines the choice of crop types [27]. In previous research, China generally chooses maize-wheat, pea, soybean and peanut intercropping, India generally chooses wheat-chickpea, coriander, radish, beet root, onion and mungbean, the United States generally chooses peanut-watermelon or maize-cowpea or blue panic-alfalfa intercropping, and Australia generally chooses lentil-mustard intercropping [15]. The accumulated temperature during the growing season of crops in the Yellow River irrigation area of Ningxia, which is greater than or equal to 10°C, is 3200–3400°C [28]. According to the photothermal conditions and genetic characteristics of crops, after wheat harvest, sorghum-sudangrass hybrid, soil sunflower or maize silage were replanted, and after cabbage harvest, maize silage was replanted. Although these crops cannot fully mature, but they can meet the requirements for harvesting silage forage. In intercropping planting mode, due to the complementary effects of edge row effect and water, fertilizer, light and heat resources, the crop yield for intercropping is generally higher than that of monoculture with the same land area [29]. Many studies have demonstrated that intercropping wheat and maize can markedly enhance crop yields in comparison to monoculture practices [23,30]. However, it is widely believed that wheat yield will increase while maize yield will decrease [31]. The results of this study indicate that when wheat and maize silage are intercropped, both show increased yields compared to monoculture (Table 2). This is because the distance between the wheat and maize silage rows was expanded to 30 cm during sowing, which first minimized the stress effect of wheat on the early growth of maize silage. The LER of the wheat-maize intercropping system is influenced by the planting ratio of the two crops, an intercropping mode consisting of six wheat rows and two maize rows (6:2), as well as additional row configurations (8:2, 6:3), exhibited LERs ranging from 1.18 to 1.30 [32]. Other intercropping systems, such as maize-soybean and soybean-wheat intercropping systems, also exhibit notable intercropping advantages, with LER values ranging between 1.02 and 1.17 [29]. However, in this study, the LER of each mode ranged from 1.76 to 2.3 in two years. This is mainly because the planting modes in this study involves various forms such as intercropping and multiple cropping, changing from two crops to three or more, thus greatly improving the LER of each diversified planting mode.

Under the intercropping mode between two crops, due to the mutual influence between the aboveground and underground parts, there is generally a competitive relationship between the two crops in terms of growth resources such as water, fertilizer, gas, and heat [33], especially soil moisture is usually the key to crop growth [34]. In the traditional maize-wheat intercropping system, due to the early sowing of wheat, it often has an advantage in intercropping during growth. Although maize may recover to a certain extent in the later stage, it still remains at a disadvantage in intercropping [29]. However, in this study, we appropriately increased the distance between the wheat edge row and the maize silage edge row to reduce

the resource competition of wheat for early growth of corn, and through appropriate water and fertilizer management in the later stage, the growth of corn was restored. After calculation, it was found that maize silage had a competitive advantage over wheat at maturity (Fig 3). There is relatively little research on the intercropping of maize silage and sorghum-sudangrass hybrid. When these two crops are intercropped, corn has a significant intercropping advantage because sorghum-sudangrass hybrid is replanted after wheat harvest, and corn is in the tasseling silking stage, with a larger plant height and developed root system, which can absorb more water and fertilizer resources. In the intercropping system between maize and sunflowers, sunflowers are generally the competitive advantage crop [35]. In this study, soil sunflowers were found to have a competitive advantage in the intercropping system of maize silage and soil sunflowers after wheat harvest, which is consistent with previous research results. However, in T3 treatment, maize silage has a competitive advantage, which may be due to the earlier harvest of cabbage than wheat in the previous crop, the earlier time of maize silage replanting, and the significant growth advantage of maize silage when sunflowers emerge (Fig 3).

The intercropping system can achieve diversified planting according to the market, which increases the ability to resist market risks compared to sole planting. Due to the different types of crops planted, the economic benefits also vary greatly, ranging from 10,000 to 95,000 yuan/ha [14,36]. The average net benefit of T1-T3 treatment in this study ranges from 61,900 to 85,000 yuan/ha over two years (Table 3). It can be seen that under the intercropping multiple cropping planting mode, not only can the output efficiency of land be improved, but also the economic benefits of agricultural production can be enhanced. However, it is necessary to choose a reasonable intercropping planting mode based on local planting habits, market demand, and production conditions.

## Conclusion

This study compared the biological yield and economic benefits of intercropping and multiple cropping diversified planting modes, and analyzed the optimal planting mode of intercropping and multiple cropping in the Yellow River irrigation area of Ningxia. The results showed that due to the implementation of multiple crop planting on the same land area within a year, as well as the mixed planting of various crops such as vegetables, forage, and grains, not only did the grain yield greatly increase, but also the harvest of vegetables and forage was achieved, supporting regional food security and the rapid development of animal husbandry.

## Author contributions

**Conceptualization:** Jing Chen.

**Data curation:** Jianxin Jin.

**Formal analysis:** Jianxin Jin.

**Funding acquisition:** Na Zhang.

**Investigation:** Jianxin Jin.

**Methodology:** Jing Chen.

**Project administration:** Jing Chen.

**Resources:** Na Zhang.

**Software:** Na Zhang, Jianxin Jin.

**Supervision:** Jing Chen.

**Validation:** Na Zhang.

**Visualization:** Na Zhang.

**Writing – original draft:** Na Zhang.

**Writing – review & editing:** Na Zhang.

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
