## [Decision Letter · Decision Letter 0]

25 Nov 2024

PONE-D-24-46367Water production efficiency and economic benefits under diversified planting modes of intercropping-multiple cropping in arid regionsPLOS ONE

Dear Dr. Jing Chen,

Thank you for submitting your manuscript to PLOS ONE. After careful consideration, we feel that it has merit but does not fully meet PLOS ONE’s publication criteria as it currently stands. Therefore, we invite you to submit a revised version of the manuscript that addresses the points raised during the review process.

The reviewers feel   that:

In the introduction, is not sufficient delving into background review to justify importance of the study (more literature required, preferably no older than 5 years. In Methods, although one reviewer commended the general model applied in materials and methods, the other thought that there would better be need to Include a conceptual or diagrammatic framework outlining the experimental protocol. Also the study design needs to be more clearly out-layed and the research objectives/questions presented more clearly to address gaps in research that these questions seek to address. There is also need for a clear (set of) hypotheses, including the authors’ own expectations of what the findings might be (based on that/those hypotheses. For units, the authors need to settle for either metric (such as m) or hectares throughout

In the results, there is need to Include a clear outline of the merits of the range of experimental treatments i.e. show which treatments ranked higher than which

In the discussion, the authors need to adopt a logical sequence approach in discussing the results obtained, laying the greatest emphasis on the key findings, and also putting the results in the context of existing literature (in other words comparing with other studies elsewhere).

With regards to references, the authors should closely check for compliance with PLOS ONE guidelines

Ensure all cited references are listed at the end and vice versa

Avoid duplicating references in the  reference listing

Please also take note of the PDF attachment with a reviewer's additional specific comments

We look forward to receiving your revised manuscript.

Kind regards,

Nickson E. Otieno

Academic Editor

PLOS ONE

Journal Requirements:

“This work was supported by the National Key Research & Development Program of China (2021YFD1900600, 2022YFG1900205); the Key Research & Development Program of Ningxia Hui Autonomous Region (2021BEF02034); the Ningxia Natural Science Foundation Project (2022AAC03725)”

4. We note that your Data Availability Statement is currently as follows: “All relevant data are within the manuscript and its Supporting Information files.”

Please confirm at this time whether or not your submission contains all raw data required to replicate the results of your study. Authors must share the “minimal data set” for their submission. PLOS defines the minimal data set to consist of the data required to replicate all study findings reported in the article, as well as related metadata and methods (https://journals.plos.org/plosone/s/data-availability#loc-minimal-data-set-definition ).

If your submission does not contain these data, please either upload them as Supporting Information files or deposit them to a stable, public repository and provide us with the relevant URLs, DOIs, or accession numbers. For a list of recommended repositories, please see https://journals.plos.org/plosone/s/recommended-repositories .

If there are ethical or legal restrictions on sharing a de-identified data set, please explain them in detail (e.g., data contain potentially sensitive information, data are owned by a third-party organization, etc.) and who has imposed them (e.g., an ethics committee). Please also provide contact information for a data access committee, ethics committee, or other institutional body to which data requests may be sent. If data are owned by a third party, please indicate how others may request data access."

5. We note that Figure 1 in your submission contain map images which may be copyrighted. All PLOS content is published under the Creative Commons Attribution License (CC BY 4.0), which means that the manuscript, images, and Supporting Information files will be freely available online, and any third party is permitted to access, download, copy, distribute, and use these materials in any way, even commercially, with proper attribution. For these reasons, we cannot publish previously copyrighted maps or satellite images created using proprietary data, such as Google software (Google Maps, Street View, and Earth). For more information, see our copyright guidelines: http://journals.plos.org/plosone/s/licenses-and-copyright  

Reviewers' comments:

Reviewer's Responses to Questions

**Comments to the Author**

1. Is the manuscript technically sound, and do the data support the conclusions?

Reviewer #1: Yes

Reviewer #2: Yes

2. Has the statistical analysis been performed appropriately and rigorously? 

Reviewer #1: Yes

Reviewer #2: Yes

3. Have the authors made all data underlying the findings in their manuscript fully available?

Reviewer #1: Yes

Reviewer #2: Yes

4. Is the manuscript presented in an intelligible fashion and written in standard English?

Reviewer #1: No

Reviewer #2: Yes

5. Review Comments to the Author

Reviewer #1: Specific Comments

Abstract

The abstract needs to be revised extensively. Please find comments in attached pdf file.

Introduction

The introduction is not providing enough information on the research background and research gap. The authors are advised to add more relevant literature to introduce the knowledge gap.

Materials and Methods

The authors are advised to add clear graphics to show configurations used in the experiment as the textual content is unclear for reader.

Results

I strongly recommend the authors to write results with exact significant values and then calculate percentage differences between best and worst treatments. And please recheck the results as author wrote PLER in the result and discussion section while in the associated table it is written PLEW. Please review the whole paper for consistency of terminologies and unites.

Discussion

2. I think authors should rethink what they write in the first paragraph and only summarize the main findings in view of the research questions. After this, authors can explore different aspects of the work in subsequent paragraphs and explain how their findings expand the envelope of knowledge, but first of all, authors simply need to state the main results without discussing their why and how or the relationships to the literature. First of all, the reader needs a clear statement on what the study found. Moreover, it is suggested to discuss the main results in a logical way.

Authors Read following paper on soil water balance and may include in citation.

Yi, J., Li, H., Zhao, Y., Shao, M., Zhang, H.,... Liu, M. (2022). Assessing soil water balance to optimize irrigation schedules of flood-irrigated maize fields with different cultivation histories in the arid region. Agricultural Water Management, 265, 107543. doi: https://doi.org/10.1016/j.agwat.2022.107543

Reviewer #2: 1. Line 24: 16.81 RMB/m3. In fact, I would advise the author to convert to net benifit per hectare.

2. Line 28: m2? In the previous paragraph it is m3

3. Line 53: (LI et al., 2021). Misquoted edit

4. The Introduction section lacks references to the latest relevant research, e.g. Wang et al., 2024. https://doi.org/10.1016/j.resconrec.2024.107898

5. Line 75: In wheat-maize intercropping system. You can refences the following research:

https://doi.org/10.1016/j.catena.2023.107247

6. Lines 79-80: A vague question. I recommend that authors clarify their research objectives, and to come up with a reasonable presupposition.

7. Line 87: This model can be used in many irrigation districts around the world, not just in experimental area.

8. M&M: Very comprehensive content. The test was done very beautifully.

9. Line 135-156: These formulas need to be supplied with a citation.

10. Line 161: Intercropping

11. Please replace all statistical units with the unit hectare (ha) instead of 667 m2, as this is not internationally common.

12. Table 2: Add line spacing or spaces so that each treatment is clearly distinguishable to the reader

13. Line 212: Interspecies

14. 14. In fact, heat is the key determinant of polyculture. The authors were able to grow multiple crops in a single growing season because sufficient heat was available. Moreover, some crops are heat-loving crops and some are cool-loving. These differences in growth habits between crops lead to complementary ecological niches, which in turn determine yields. Please add relevant discussions：

Chai et al., 2021, PNAS

15. Duplicates appear in the references list, e.g. Wang 2022

6. PLOS authors have the option to publish the peer review history of their article (what does this mean? ). If published, this will include your full peer review and any attached files.

**Do you want your identity to be public for this peer review?** For information about this choice, including consent withdrawal, please see our Privacy Policy .

Reviewer #1: No

Reviewer #2: No

---

## [Author Response · Author response to Decision Letter 1]

21 Dec 2024

[December 16, 2024]

Dear editor:

We have resubmitted the revised manuscript of “Water production efficiency and economic benefits under diversified planting modes of intercropping-multiple cropping in arid regions” to the PLoS One editorial department. We would like to express our sincere gratitude to the editorial department and reviewers for their invaluable feedback on the manuscript, which has significantly contributed to enhancing its overall quality. Upon thoroughly reviewing the comments, we have diligently implemented comprehensive revisions to the manuscript. The updated content is clearly marked in blue font within the document, and each review comment has been addressed individually in the accompanying revision notes. Please find the revised manuscript and detailed revision instructions attached. Thank you for your consideration. We look forward to hearing from you.

Additional instructions:

(1)The funders had no role in study design, data collection and analysis, decision to publish, or preparation of the manuscript.

(2)The original data of this manuscript is submitted to the editorial department in the form of an attachment.

(3)The original map in Figure 1 is a GIS image. To avoid potential copyright issues, the revised manuscript has removed the map graphics and only retained photos of the experimental site.

Sincerely,

Jing Chen

Hohai University

Nanjing 211100, China Telephone: +86-157-6959-1162

Email: 2156640980@qq.com

---

## [Decision Letter · Decision Letter 1]

19 Jan 2025

Water production efficiency and economic benefits under diversified planting modes of intercropping-multiple cropping in arid regions

PONE-D-24-46367R1

Dear Dr. Zhang,

We’re pleased to inform you that your manuscript has been judged scientifically suitable for publication and will be formally accepted for publication once it meets all outstanding technical requirements.

Kind regards,

Nickson E. Otieno

Editor

PLOS ONE

Additional Editor Comments (optional):

In addition to line numbering, all pages must be numbered sequentiallyReplace ‘interspecies’ with ‘interspecific’ throughout the textMust leave a line space(skip a line)  between end of one section and title heeding or subheading  of the next section throughout the article.Also leave spaces between tables and paragraphs of between paragraphs and figuresProvide a lot more details in table captions and figure legends to include tests from which they are generated, and indicate standard errors or standard deviations and sample/population sizes (n or N) of the data they come from. Currently you have only a few words  un table captions and figure legends. This is not acceptablein Table 1, 2 and 3, the text in the whole column titled ‘Crop’ should be left-justified (do not center the text)as was pointed oout at revision 1 and again at revision 2 in review comments, remove all duplicate references listed, and ensure all text-cited references are listed and all listed references are text-cited

Reviewers' comments:

Reviewer's Responses to Questions

**Comments to the Author**

1. If the authors have adequately addressed your comments raised in a previous round of review and you feel that this manuscript is now acceptable for publication, you may indicate that here to bypass the “Comments to the Author” section, enter your conflict of interest statement in the “Confidential to Editor” section, and submit your "Accept" recommendation.

Reviewer #2: All comments have been addressed

2. Is the manuscript technically sound, and do the data support the conclusions?

Reviewer #2: Yes

3. Has the statistical analysis been performed appropriately and rigorously? 

Reviewer #2: Yes

4. Have the authors made all data underlying the findings in their manuscript fully available?

Reviewer #2: Yes

5. Is the manuscript presented in an intelligible fashion and written in standard English?

Reviewer #2: Yes

6. Review Comments to the Author

Reviewer #2: The author has responded well to my concerns. I think the current version can be published directly.

7. PLOS authors have the option to publish the peer review history of their article (what does this mean? ). If published, this will include your full peer review and any attached files.

**Do you want your identity to be public for this peer review?** For information about this choice, including consent withdrawal, please see our Privacy Policy .

Reviewer #2: No

---

## [Editor Report · Acceptance letter]

PONE-D-24-46367R1

PLOS ONE

Dear Dr. Zhang,

I'm pleased to inform you that your manuscript has been deemed suitable for publication in PLOS ONE. Congratulations! Your manuscript is now being handed over to our production team.

Kind regards,

on behalf of

Dr. Nickson E. Otieno

Academic Editor

PLOS ONE